# ARRDC4 and UBXN1: Novel Target Genes Correlated with Prostate Cancer Gleason Score

**DOI:** 10.3390/cancers13205209

**Published:** 2021-10-17

**Authors:** Jong Jin Oh, Jin-Nyoung Ho, Seok-Soo Byun

**Affiliations:** 1Department of Urology, Seoul National University Bundang Hospital, Seongnam-si 463-707, Korea; bebsuzzang@naver.com (J.J.O.); 97875@snubh.org (J.-N.H.); 2Department of Urology, Seoul National University College of Medicine, Seoul 03080, Korea

**Keywords:** prostate cancer, exome array

## Abstract

**Simple Summary:**

Prostate cancer (PCa) is the second most diagnosed malignancy in men. PCa is a heterogeneous disease, with the clinical presentation ranging from localized and indolent to a rapidly progressing lethal metastatic disease, therefore, we needed the predictor to diagnosis of aggressive PCa. The most important predictor of PCa outcomes has been demonstrated to be the histological Gleason score which was confirmed after prostate biopsy. In this study, we found that germline variants of ARRDC4 and UBXN1 could affect the prostate cancer Gleason score, which could be a potential marker to select aggressive PCa.

**Abstract:**

To investigate potential markers of the prostate cancer (PCa) Gleason score (GS), genetic arrays in 841 PCa patients were conducted followed by functional validation in PCa cell lines. A total of 841 PCa patients who received radical prostatectomy (RP) from November 2003 to July 2019 were enrolled. HumanExome BeadChip 12v1-1 (Illumina, Inc.; San Diego, CA, USA) exomic arrays were performed on RP tissue samples. Unconditional logistic regression was used to calculate odds ratios to generate estimates of the relative risk of pathologic GS (≥8); SNPs with the highest association were selected and validated using PCa cell lines (PC3, LNCaP, 22Rv1 and DU145). Following transfection with target-gene siRNA, assays for cell viability, wound healing, and transwell invasion were performed. Mean age of enrolled subjects was 66.34 years and median PSA was 8.43 ng/mL. After RP, 122 patients (14.5%) had pathological Gleason scores ≥8. The results from genotyping with 242,186 SNPs by exomic array revealed that 4 SNPs (rs200944490, rs117555780, rs34625170, and rs61754877) were significantly associated with high pathological GS (≥8) within cut-off level to *p* < 10^−5^. The most highly associated rs200944490 in ARRDC4 (*p* = 1.39 × 10^−6^) and rs117555780 in UBXN1 (*p* = 2.92 × 10^−5^) were selected for further validation. The knockdown of UBXN1 and ARRDC4 led to significantly reduced cell proliferation and suppressed migration and invasiveness in PCa cell lines. Epithelial mesenchymal transition (EMT) markers were significantly down-regulated in si-ARRDC4 and si-UBXN1-transfected cells. The expression levels of PI3K-phosphorylation and Akt phosphorylation and NF-κB were also suppressed following knockdown of UBXN1 and ARRDC4. The rs200944490 (ARRDC4) and rs117555780 (UBXN1) were identified as candidate markers predictive of PCa Gleason score which is strongly associated with cancer aggressiveness. Additional validation in future studies is warranted.

## 1. Introduction

Prostate cancer (PCa) is the second most diagnosed malignancy in men and the sixth leading cause of cancer mortality among males worldwide [1]. In South Korea, the incidence of PCa has increased significantly in recent years, partly due to the rise in average life expectancy, westernized dietary habits, and increased awareness of PCa screening; between 2007 and 2013, the incidence of PCa doubled (from 5516 per year to 10,855 per year), while its prevalence in Korean men tripled (from 18,830 to 51,411) [2].

PCa is a heterogeneous disease, with the clinical presentation ranging from localized and indolent to a rapidly progressing lethal metastatic disease [3,4,5,6]. Although most diagnoses involve organ-confined disease, long-term oncological outcomes can vary greatly [7,8,9]. The most important predictor of PCa outcomes has been demonstrated to be the histological Gleason score. The Gleason grading system, based upon architectural features of prostate cancer cells, is the most widely used histological grading method for prostatic adenocarcinoma. The Gleason score closely correlates with clinical features and serves as an important prognostic index [10]. PCa with Gleason scores of 6 and 8 represent low-risk and high-risk PCa with 5-year progression-free survival rates of 93% and 72.1%, respectively [11,12].

Treatment plans tend to be developed using: (1) prostate-specific antigen (PSA) levels, (2) clinical stage with or without magnetic resonance imaging, and (3) biopsy Gleason scores. However, there has been discrepancy between biopsy Gleason score and final pathologic Gleason score. For instance, one report demonstrated a 25.5% discrepancy between biopsy Gleason score and pathologic Gleason score after radical prostatectomy (RP) [13]. Therefore, additional parameters which can increase the accuracy of biopsy Gleason scores would be beneficial. In this study, the authors sought to identify and characterize genetic alterations which correlated with Gleason scores using an exomic array approach (step 1) followed by validation of these target genes using PCa cell lines (step 2).

## 2. Material and Methods

### 2.1. Ethics Statement

This study was approved by our institutional review board (Seoul National University Bundang Hospital’s Institutional Review Board; IRB number, B-1312/232-302) and followed the rules stated in the Declaration of Helsinki. All participants provided written informed consent.

### 2.2. Study Population

Between November 2003 and July 2019, 1002 patients with PCa were screened for inclusion in this study. Among them, 841 patients who received radical prostatectomy (RP) and had confirmative pathological Gleason score were finally enrolled. All RP specimens were analyzed by a single genitourinary pathologist.

### 2.3. Genotyping and Quality Control

Using the HumanExome BeadChip 12v1-1 (Illumina, Inc.; San Diego, CA, USA), which includes 242,901 markers focused on protein-altering variants, exomic array was performed. Details of the SNPs and selection strategies can be found at the exome array design web page (http://genome.sph.umich.edu/wiki/Exome_Chip_Design). Genotyping was performed using Illumina’s GenTrain version 2.0 clustering algorithm with the GenomeStudio software (version 2011.1) (Illumina Inc., San Diego, CA). Cluster boundaries were determined using Illumina’s standard cluster file. After additional visual inspection of SNPs with call rates of <0.99 and SNPs with minor allele frequencies of <0.002, of the 242,901 attempted markers, 242,186 (99.71%) were successfully genotyped with call rates of 495% (average call rate 99.98%). In total, 1001 of 1002 (99.9%) individuals were successfully genotyped (call rate >98%). For the 242,186 SNPs that passed quality control, genotype concordance among the 104 blind duplicate sample pairs was 99.998%. One individual per pair of six known twin pairs and six unknown duplicates were excluded. We performed principal components (PCs) analysis twice, once excluding HapMap samples to identify population outliers and a second time including HapMap samples to interpret outliers. To avoid artifacts due to family relatedness, we computed the PCs using SNP loadings estimated from a subset of 7304 individuals who were determined not to be close relatives. We defined close relatives as an estimated genome-wide identical-by-descent proportion of alleles shared of >0.10. We estimated identical-by-descent sharing using PLINK’s “-genome” option 38 and performed PCs analysis using SMARTPCA37 on a linkage disequilibrium-pruned set of 22,464 autosomal SNPs. These were obtained by removing large-scale, high-linkage disequilibrium (LD) regions, SNPs with a minimum allele frequency <0.01, or SNPs with a Hardy–Weinberg equilibrium *p* < 10^−6^ and performing LD pruning using the PLINK option “–indep-pairwise 50 5 0.2”.

### 2.4. Exome Array SNP Analysis

SNP genotype frequencies were examined for Hardy–Weinberg equilibrium using the chi-squared test, and all were found to be consistent (*p* > 0.05). The data were analyzed using an unconditional logistic regression to calculate an odds ratio as an estimate of the relative risk of PCa pathologic Gleason score (≥8) associated with the SNP genotypes. To determine the association between genotype and haplotype distributions, a logistic analysis was performed controlling for age (continuous value) as a covariate to eliminate or reduce any confounding factors that could influence the findings. Lewontin’s D’ (|D’|) and the LD coefficient *r*^2^ were examined to measure LD between all pairs of biallelic loci [14]. The haplotypes were inferred from the successfully genotyped SNPs with the PHASE algorithm version 2.0, using SAS version 9.1 (SAS Inc.; Cary, NC, USA). The effective number of independent marker loci was calculated to correct for multiple testing using SNPSpD software (http://www.genepi.qimr.edu.au/general/daleN/SNPSpD/), which is based on the spectral decomposition of matrices of pairwise LDs between SNPs [15].

### 2.5. Cell Culture and Reagents

The human prostate cancer cell lines PC3, LNCaP, 22Rv1, and DU145 were purchased from American Type Cell Culture Collection (ATCC, Rockville, MD, USA). PC3, LNCaP, and 22Rv1 cells were maintained in RPMI 1640 medium containing 10% fetal bovine serum (FBS) and 100 units/mL penicillin and 100 mg/mL streptomycin (Gibco, Thermo Fisher Scientific Inc. Grand Island, NY, USA). DU145 cells were routinely grown in DMEM medium supplemented with 10% FBS.

### 2.6. UBXN1-siNRA and ARRDC4-siRNA Transfection

Cells were seeded in a 60 mm dish, cultured to a confluency of 70% and transiently transfected with UBX domain-containing protein 1 (UBXN1)-small interfering RNA (siRNA), arrestin domain containing 4 (ARRDC4)-siRNA, and negative control-siRNA (Ambion, Austin, TX, USA) at 20 nM final concentration using Lipo iMAX reagent (Invitrogen, Grand Island, NY, USA) in Opti-MEM (Gibco, Thermo Fisher Scientific Inc. Waltham, MA, USA).

### 2.7. Quantitative Real-Time PCR (qRT-PCR)

The efficiency of gene knockdown was measured 48 h after transfection by quantitative real-time PCR. Total RNA was extracted from transfected cells using the RNeasy Mini kit (Qiagen, Maryland, MD, USA). cDNA syntheses were performed in 20 µL of reaction reagent plus 1 µg RNA using the iScript cDNA synthesis kit (Bio-Rad, Hercules, CA, USA). Using the Power SYBR green PCR master mix (Applied Biosystems, Warrington, UK), PCR reactions were performed with initial step at 95 °C for 10 min; and 40 cycles of denaturation (95 °C) for 5 sec and annealing (58 °C) for 30; extension was conducted at 72 °C for 30 s using ViiA7 system (Applied Biosystems). GAPDH was used as the loading control (Appendix A).

### 2.8. Cell Counting Kit-8 (CCK-8) Assay

Following transfection, cell viability was evaluated by CCK-8 assay (Dojindo Molecular Technologies, Gaithersburg, MD, USA). Cells were seeded in 96-well plates and transfected with control, UBXN, or ARRDC4 siRNAs. After 24 and 48 h of incubation, viability of transfected cells was measured by CCK-8 assay. CCK-8 solutions were added to wells followed by incubation for 4 h. The absorbance value at 450 nm was detected using a microplate reader (Molecular Devices, Sunnyvale, CA, USA).

### 2.9. Wound Healing Assay

To quantify cell migration, wound healing assays were performed. Cells were cultured into six-well plates to 90% confluence and scratched with 200 µL sterile tips followed by incubation for 24 h or 48 h; wound healing was observed and photographed with a microscope. The rate of open wound area was measured and calculated.

### 2.10. Transwell Invasion Assay

Invasion assays were performed using transwell approach (6.5 mm insert, 8.0 µm pore size; Corning, NY, USA); the upper chamber (Corning Costar, New York, NY, USA) was precoated with Matrigel (BD Bioscience, San Jose, CA, USA) according to the manufacturer’s protocols before 5 × 10^4^ cells in serum-free DMEM were added to the chamber, and the lower chamber was loaded with culture media containing 10% FBS. After 48 h of incubation, non-invaded cells were removed with cotton swabs, and invaded cells were fixed and stained using Diff Quik (Sysmex, Kobe, Japan), and counted under a microscope.

### 2.11. Western Blot

Total cellular proteins were extracted with RIPA buffer, containing of 50 mM Tris-HCl (pH 8.0), 150 mM sodium chloride, 1.0% NP-40, 0.5% sodium deoxycholate, 0.1% SDS, and 1 mM phenylmethylsulfonyl fluoride (PMSF) on ice for 1 h. Protein concentration of cell lysates were determined by Bradford protein assay (Bio-Rad). Equal amounts of protein were loaded in 8–12% SDS-polyacrylamide gel electrophoresis and transferred to a PVDF membrane (Milipore, Billerica, MA, USA). Membranes were blocked with 5% skim milk and incubated with primary antibodies (PARP, cleaved caspase-3, -8, -9, cytochrome c, bcl-2, bad, E-cadherin, vimentin, p-PI3K, PI3K, p-Akt, Akt, NF-κB p65; diluted 1:1000; Cell Signaling Technology Inc., Beverly, MA, USA) overnight at 4 °C. Finally, membranes were incubated with secondary antibody (diluted 1:5000~1:10,000). The blots were visualized with the ECL reagent (Pierce, Rockford, IL, USA).

### 2.12. Statistical Analysis

The data were expressed as mean ± standard deviation. The statistical analysis was conducted using SPSS soft package (IBM SPSS statistics 20) (SPSS Inc., Chicago, IL, USA). *p*-value < 0.05 was considered to indicate a significant difference.

## 3. Results

### 3.1. Baseline Characteristics

As shown in Table 1, among 841 PCa patients, mean age was 66.34 years and median PSA was 8.43 ng/mL. After RP, 265 (31.5%) extracapsular extension (ECE) and 80 (9.5%) seminal vesicle invasion (SVI) and 249 positive surgical margin (PSM) was found in total cohorts. Of all patients, 122 (14.5%) had pathological Gleason score ≥8, 190 patients (22.6%) and 78 patients (9.3%) had pathological T3a stage and T3b stage, respectively.

### 3.2. Genotyping

The genotype frequencies in all patients according to pathological Gleason score (cut-off—Gleason score 8) were analyzed using a logistic regression model, and the data are presented in Figure 1 (Manhattan plot). The results from genotyping with 242,186 SNPs by exomic array revealed that 4 SNPs (rs200944490 in Chromosome (Chr) 15, rs117555780 in Chr 11, rs34625170 in Chr 1, and rs61754877 in Chr 3) were significantly associated with high pathological Gleason score (≥8) after RP within cut-off level to *p* < 10^−5^ (Table 2). All four SNPs were strongly positively associated with high Gleason scores; odds ratios ranged from 1.759 to 6.459.

The most highly associated rs200944490 (*p* = 1.39 × 10^−6^) and rs117555780 (*p* = 2.92 × 10^−5^), which was near located within the ARRDC4 and UBXN1 genes, respectively, were selected for further validation using human PCa cell lines.

### 3.3. Expression of UBXN and ARRDC4 Following siRNA Transfection in PCa Cells

siRNAs targeting UBXN1 and ARRDC4 were used to knockdown the expression of these genes in prostate cancer cells. After transfection with UBXN1 and ARRDC4-siRNA in various PCa cell lines (PC3, DU145, LNCaP, and 22Rv1), real-time PCR was conducted to validate the effect of siRNA on the expression levels of endogenous UBXN1 and ARRDC4. As expected, UBXN1 and ARRDC4-siRNA significantly decreased endogenous expression of UBXN1 and ARRDC4 levels compared to the negative control-siRNA transfected cells (Figure 2), demonstrating transfection effectiveness and on-target action of these siRNAs.

### 3.4. Knockdown of UBXN1 and ARRDC4 Inhibited the Proliferation of Prostate Cancer Cells

Prostate cancer cell proliferation was measured by CCK-8 assay. As shown in Figure 3, human PCa cell lines presented different degrees of sensitivity to knockdown of UBXN1 and ARRDC4 on cell proliferation. Transfection with UBXN1-siRNA led to significantly reduced cell proliferation in PC3, DU145, LNCaP, and 22Rv1 cells in comparison with the negative control-siRNA transfected cells. Similarly, low levels of ARRDC4 following siRNA transfection markedly inhibited proliferation of DU145 and LNCaP cells compared with control-siRNA transfected cells. Cell proliferation of PC3 and 22Rv1 cells also decreased following knockdown of ARRDC4, however, this change was not significant compared with control cells. Collectively, these results indicated that knockdown of UBXN1 could suppress proliferation in PCa cell lines. The lower expression levels of ARRDC4 also seem to impact proliferation in some of the PCa cell lines tested.

### 3.5. Knockdown of UBXN1 and ARRDC4 Inhibited Migration and Invasion of PCa Cells

The roles of UBXN1 and ARRDC4 in migration and invasion capacity of human PCa cell lines were characterized. As illustrated in Figure 4A–D, siRNA-induced knockdown of UBXN1 significantly suppressed the migration of PC3, DU145, LNCaP, and 22Rv1 cells compared with the negative control-siRNA transfected cells. The migration of ARRDC4-siRNA-transfected PC3 and 22Rv1 was also decreased. However, no significant difference in the ARRDC4-siRNA transfected and negative control-siRNA transfected cells was demonstrated. Knockdown of ARRDC4 does not affect migration of DU145 and LNCaP prostate cancer cells.

Similarly, the invasiveness of PCa cells transfected with UBXN1-siRNA remarkably decreased compared with the negative control-siRNA-transfected cells. The inhibition rate of invasiveness in UBXN1-siRNA transfected PC3 cells was 84.17%, for DU145 cell it was 70.41%, for LNCaP cell it was 61.17%, and for 22Rv1 cell it was 44.07% (Figure 4E–H). Invasiveness of LNCaP and 22Rv1 cells transfected with ARRDC4-siRNA was significantly reduced by 43.24% and 43.9%, respectively, when compared to negative control-siRNA-transfected cells. Although the invasiveness of PC3 cells transfected with ARRDC4-siRNA decreased, the difference compared with control cells was not significant. These findings strongly support that knockdown of UBXN1 could suppress prostate cancer cell migration and invasion.

### 3.6. Knockdown of UBXN1 and ARRDC4 Reduced Apoptosis and EMT

The apoptosis-related genes, cleaved caspase-3, -8, -9, fragmented PARP, cytochrome c, and bad were quantified by Western blot. The levels of cleaved caspases-3, -8, -9, fragmented PARP, cytochrome c, and bad were obviously increased following UBXN1-siRNA transfection compared to the negative control (Figure 5). However, the expression of the apoptosis-associated genes in PCa cells transfected with ARRDC4-siRNA were slightly enhanced or not changed. Since epithelial-to-mesenchymal transition (EMT) actively relates to the invasion and metastasis of cancer cells [16,17], the expression of EMT markers such as vimentin and E-cadherin in PCa cells following knockdown of UBXN1 and ARRDC4 was characterized. Reduced expression of UBXN1 and ARRDC4 in PCa cells resulted in down-regulation of vimentin (an important mesenchymal marker), and upregulation of E-cadherin (commonly used epithelial marker) compared with negative control-siRNA-transfected cells.

### 3.7. Knockdown of UBXN1 and ARRDC4 Suppressed PI3K/Akt/NF-κB Pathway

The PI3K/Akt pathway is one of the most prominent alternate pathways in human cancer and is elevated in a high proportion of PCa patients. The PI3K/Akt signaling pathway is involved in cell growth, differentiation, migration, metastasis, proliferation in many human cancers [18,19]. Furthermore, the PI3K/Akt pathway, which is upstream of NF-κB is mediated by NF-κB; activation of NF-κB results in the acquisition of the EMT phenotype and lower rates of apoptosis [20,21]. Therefore, to further analyze the possible mechanism of UBXN1 and ARRDC4 in invasion, apoptosis and EMT of prostate cancer cells, Western blotting was used to characterize the protein levels of p-PI3K, PI3K, p-Akt, Akt, and NF-κB. The expression levels of PI3K-phosphorylation and Akt phosphorylation and NF-κB were obviously suppressed following knockdown of UBXN1 and ARRDC4 in PC3, DU145, LNCaP, and 22Rv1 cells (Figure 6). Taken together, UBXN1 and ARRDC4 knockdown inhibited cell proliferation, apoptosis, migration, invasion, and EMT, changes mediated by the PI3K/Akt/NF-κB signaling pathway in PCa cells. UBXN1 appears to play an important role in cell proliferation, migration, invasion, and EMT of PCa cells.

### 3.8. Biochemical Recurrence-Free Outcomes

During a median follow-up of 58 months, 130 patients had experienced biochemical recurrence (BCR). Five-year BCR-free survival rate was achieved 24.9% in patients with variants and 69.9% in patients without (log rank test *p* < 0.001) (Figure 7)

## 4. Discussion

In this study, a two-step investigation was conducted. First, exome array analysis of post-RP tissue samples was used to identify SNPs with significant correlation to the pathologic Gleason score, a predictor of tumor aggressiveness. From this analysis, four SNPs were determined to be potential candidates for further investigation, two of which were highly associated with phenotype (rs200944490 in Chr 15 and rs117555780 in Chr 11 located within the ARRDC4 and UBXN1 genes, respectively). The second step was to further characterize the effect of these genes using various human PCa cell lines (PC3, DU145, LNCaP, and 22Rv1). Importantly, knockdown of UBXN1 and ARRDC4 resulted in the inhibition of cell proliferation, apoptosis, migration, invasion, and EMT, changes mediated via suppression of the PI3K/Akt/NF-κB signaling pathway. Based on these results, UBXN1 was demonstrated to play an important role in cell proliferation, migration, invasion, and EMT of PCa cells.

Genome-wide association studies (GWAS), in which hundreds of thousands to millions of genetic variants across the genomes of many individuals are tested to identify genotype–phenotype associations, have revolutionized the field of complex disease genetics over the past decade [22,23]. The identified associations have led to insights into the architecture of disease susceptibility and to advances in clinical care and personalized medicine [24]. However, despite clear examples demonstrating the success of GWAS, obvious limitations exist which means that not all causes of complex phenotypic diseases can be determined with GWAS alone. For this reason, functional validation is preferred to further characterize any noted phenotype–genotype correlations.

In PCa, many studies have been published on the prognostic value of the Gleason score [25,26]. While several markers (and combinations of markers) have also been identified as potential prognostic factors, to our knowledge, there are no studies describing the use of exome arrays to identify potential associations with Gleason score or pathologic Gleason Score. UBXN1, which contains a ubiquitin-associated (UBA) motif, recognizes auto-ubiquitinated BRCA1, a process that occurs through a bipartite interaction in which the UBA domain of UBXN1 binds K6-linked polyubiquitin chains conjugated to BRCA1 while the C-terminal sequences of UBXN1 bind the BRCA1/BARD1 heterodimer in a ubiquitin-independent fashion [27]. To date, no studies have identified an association between UBXN1 and PCa. However, we hypothesized that given that PCa is thought to be strongly positively associated with BRCA1 [28] and the presence of UBXN1 may act to limit the duration of BRCA1 enzymatic function, this potential correlation is logical; in this scenario, UBXN1 could affect the aggressiveness of PCa. ARRDC4 and ARRDC3 function as adapters recruiting ubiquitin-protein ligases to their specific substrates [29]. Their expression is either lost or suppressed in basal-like breast cancer and prostate cancer [30,31]. ARRDC is epigenetically silenced in cancer cells due to its promoter DNA methylation and deacetylation [32]. EMT is regarded as one of the hallmarks of cancer aggressiveness [16,17,33]. In prostate cancer, EMT changes could impact the levels of stem cells which are associated with PCa aggressiveness, and further increase Gleason patterns at the single-cell level [34,35,36]. In this study, we demonstrated changes in metastatic capacity of PCa cell lines related with the expression of UBXN1 and ARRDC4, supporting the evidence that they may serve as potential prognostic biomarkers for PCa.

This study has several limitations. First, immunohistochemical staining of selected markers was not conducted. Although 841 patients who underwent RP at a single institution were enrolled, the number of exome arrays was not sufficient. Additionally, the primary endpoint (pathological Gleason score) may have intra-observer and inter-observer bias; in this study, however, pathological reports were made by single uro-pathologist with many years of experience. The functional analysis using only one SNP could suffer from methodological limitation. To overcome this limitation, we needed polygenic risk score which aggregates the influence of SNPs in future study. Another limitation was that we only performed exome array in blood sample, without validating our results in prostatectomy tissue. Another technical limitation was that we could not perform the protein and mRNA level of ARRDC4 and UBXN1 in normal prostate, which should be resolved in a following study.

## 5. Conclusions

rs200944490 in Chr 15 and rs117555780 in Chr 11 located in the ARRDC4 and UBXN1 genes, respectively, are potential candidate markers predictive of PCa Gleason scores which strongly correlate with the aggressiveness of PCa. Further validation of the findings presented here is warranted.

## Figures and Tables

**Figure 1 cancers-13-05209-f001:**
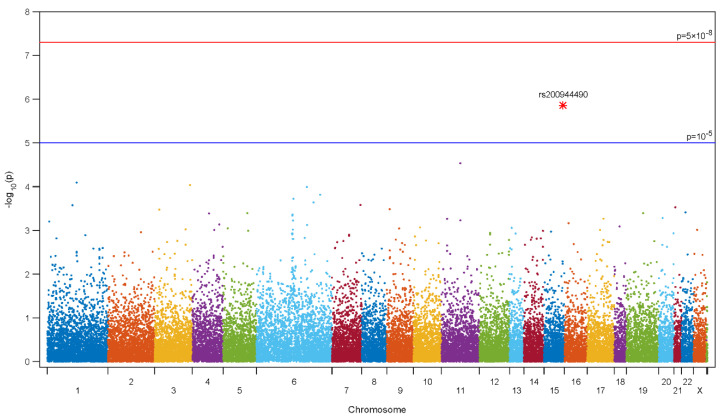
Manhattan plot. Manhattan plot of the association between high Gleason score after radical prostatectomy and 242,186 single nucleotide polymorphisms identified in patients with prostate cancer via a custom HumanExome BeadChip v1.0 (Illumina Inc. San Diego, CA, USA) (A color version of the figure is available online.)

**Figure 2 cancers-13-05209-f002:**
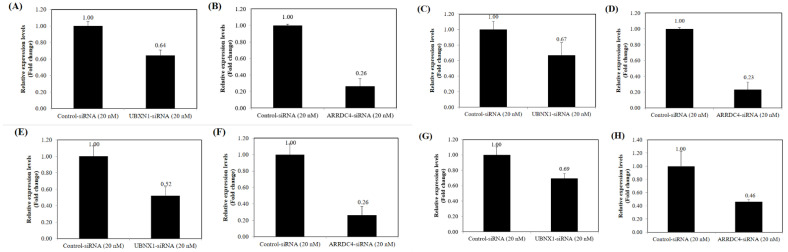
Effect siRNA transfection on UBXN1 and ARRDC4 expression. UBXN1 expression (**A**) and ARRDC4 expression in PC3 (**B**). UBXN1 expression (**C**) and ARRDC4 expression in DU145 (**D**). UBXN1 expression (**E**) and ARRDC4 expression (**F**) in LNCaP. UBXN1 expression (**G**) and ARRDC4 expression (**H**) in 22Rv1. PC3, DU145, LNCaP and 22Rv1 cells were transfected with siRNA targeted UBXN1 and ARRDC4 or its negative control. The expression of UBXN1 and ARRDC4 in these two cell lines were detected by qRT-PCR. Data are the average of at least three independent experiments.

**Figure 3 cancers-13-05209-f003:**
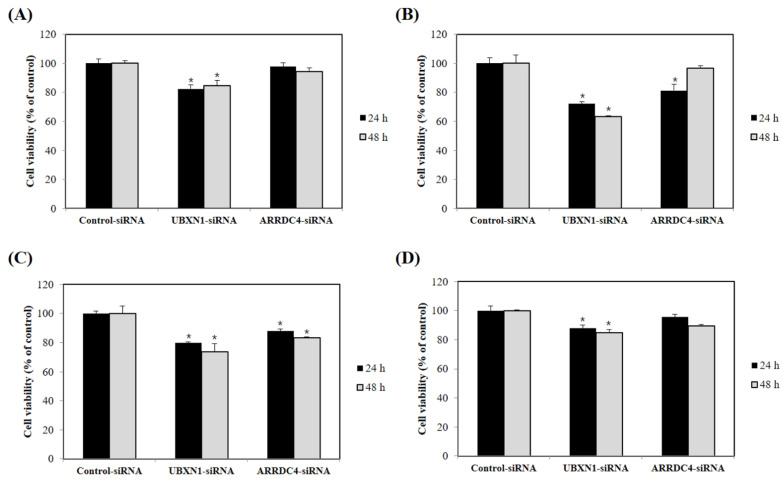
Knockdown of UBXN1 and ARRDC4 inhibits prostate cancer cell growth. PC3 (**A**), DU145 (**B**), LNCaP (**C**), and 22Rv1 cells (**D**) were transfected with siRNA targeted UBXN1 and ARRDC4 or its negative control. Cell viability was measured by CCK-8 assay. Data are the average of at least three independent experiments. * *p* < 0.05 compared to negative control.

**Figure 4 cancers-13-05209-f004:**
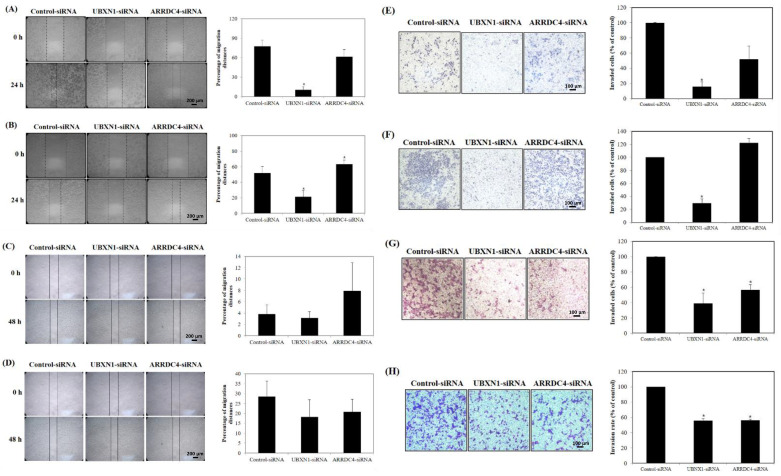
Knockdown of UBXN1 and ARRDC4 inhibits prostate cancer cell migration in PC3 (**A**), DU145 (**B**), LNCaP (**C**), and 22Rv1 cells (**D**) and invasion PC3 (**E**), DU145 (**F**), LNCaP (**G**) and 22Rv1 cells (**H**). Each cells were transfected with siRNA-targeted UBXN1 and ARRDC4 or its negative control. Cell migration and invasion were measured by wound-healing assay. Data are the average of at least three independent experiments. * *p* < 0.05 compared to negative control. Scale bar: XXX.

**Figure 5 cancers-13-05209-f005:**
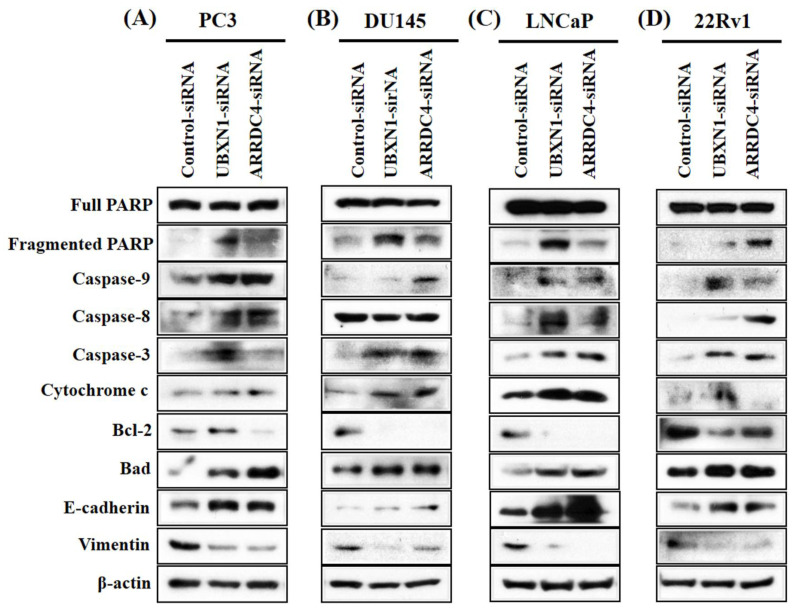
Knockdown of UBXN1 and ARRDC4 regulates crucial factors in apoptosis and EMT. PC3 (**A**), DU145 (**B**), LNCaP (**C**), and 22Rv1 cells (**D**) were transfected with siRNA-targeted UBXN1 and ARRDC4 or its negative control. The expression of apoptosis and EMT related protein were detected by Western blot analysis.

**Figure 6 cancers-13-05209-f006:**
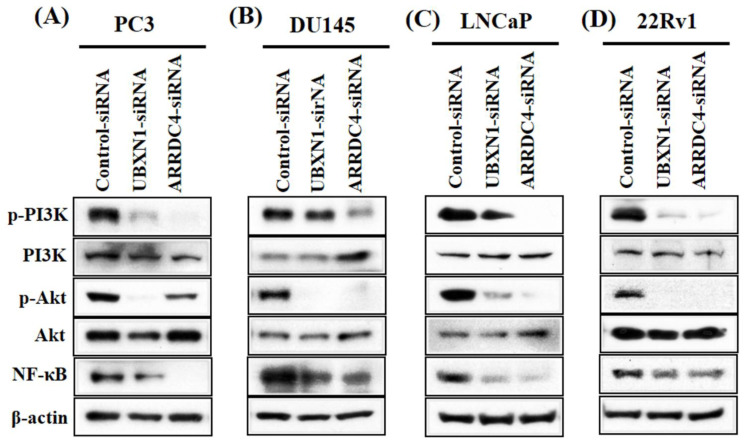
Knockdown of UBXN1 and ARRDC4 inhibits PI3/Akt/NF-kB pathway in prostate cancer cells. PC3 (**A**), DU145 (**B**), LNCaP (**C**), and 22Rv1 cells (**D**) were transfected with siRNA targeted UBXN1 and ARRDC4 or its negative control. The expression of protein was detected by Western blot analysis.

**Figure 7 cancers-13-05209-f007:**
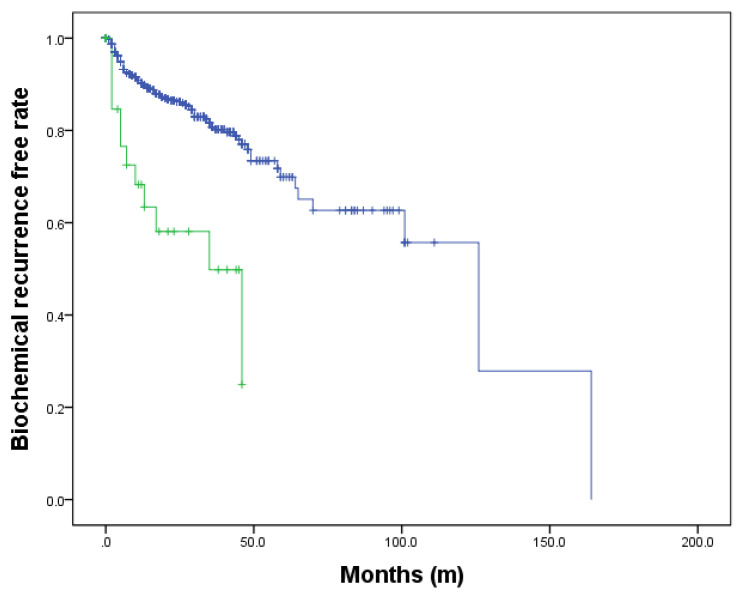
Biochemical recurrence free survival according to ARRDC4 variant.

**Table 1 cancers-13-05209-t001:** Baseline characteristics of total prostate cancer patients who underwent genetic array using exome chip.

Variables	Outcomes (Mean ± SD)
Mean age (years) ± SD	66.34 ± 6.65
Median PSA (ng/mL) ± SD	8.43 ± 19.52
Mean prostate volume (mL) ± SD	37.03 ± 16.45
Mean body mass index (kg/m^2^) ± SD	24.60 ± 8.75
Extracapsular extension (%)	265 (31.5)
Seminal vesicle invasion (%)	80 (9.5)
Positive surgical margin (%)	249 (29.5)
Lymph node invasion (%)	25 (3.0)
Pathologic stage	
pT2	565 (67.2)
pT3a	190 (22.6)
pT3b	78 (9.3)
pT4	8 (0.9)
Pathology Gleason score (%)	
6	64 (7.6)
7	655 (77.9)
8	47 (5.9)
9	75 (8.9)

PSA = prostate-specific antigen; SD = standard deviation.

**Table 2 cancers-13-05209-t002:** Logistic regression analysis of exome array to predict pathological Gleason core ≥8 after radical prostatectomy.

SNPID	Chr	Alleles	Gene	Minor Allele Frequency		OR (95% CI)	*p*-Value
				GS < 8	GS ≥ 8		
rs200944490	15	G > C	ARRDC4	0.0105	0.06087	6.459 (3.028–13.78)	1.39 × 10^−6^
rs117555780	11	G > A	UBXN1	0.0112	0.05652	4.986 (2.347–10.59)	2.92 × 10^−5^
rs34625170	1	G > A	PTGFRN	0.3153	0.4522	1.759 (1.328–2.329)	8.07 × 10^−5^
rs61754877	3	T > C	RTP2	0.08485	0.1696	2.215 (1.487–3.299)	9.18 × 10^−5^

Among 242,221 single nucleotide polymorphism, 4 SNPs significant to high Gleason score. Abbreviations: SNP = single nucleotide polymorphism; OR = odds ratio; CI: confidence interval.

## Data Availability

Restrictions apply to the availability of these data. Data are available from the authors.

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
