# Peer review of "ARRDC4 and UBXN1: Novel Target Genes Correlated with Prostate Cancer Gleason Score"

_cancers, 2021, doi:10.3390/cancers13205209_

Round 1

Reviewer 1 Report

1) General comments

The authors reported the novel genes, ARRDC4 and UBXN1, in prostate cancer. I am sure this is the first report to show an instrumental feature of these newly identified molecular in that field. And the showed an attractive profile of UBXN1 as the ubiquitin activity to BRCA1, which has been known for being one of the main gene as a hereditary breast cancer syndrome also presented as prostate cancer in men. The authors added a somewhat new prospect to the current literature; however, the study includes several points that need to be modified to qualify publication. The reviewer would like to suggest critical points to improve and increase the importance of this manuscript.

2) Specific comments:

  1. No survival outcome was not analyzed despite enough number of patients and enough period after surgery. At least, biochemical recurrence free survival should be needed. Although Gleason score has been the best predictor of treatment outcome, this pathological finding cannot be associated with biological alteration directly. For instance, nuclear grade each cell can more reflect alteration of molecular.
  2. At the first exploring exomic array analysis, the four SNPs were identified as risk of high Gleason score. Then, authors proceeded to functional analysis of ARRDC4 and UBXN1. To be general, since only one SNP should not affect molecular function, an effect of the SNPs must be analyzed or discussed.
  3. Detail of tissue preparation for exomic array was inappropriate. This methodological matter must be important and describe carefully because inappropriate tissue handling might lead to wrong results. Since Gleason score in prostate cancer is very varied, even in one patient, we should only use representative Gleason score tissue for the array. Authors should show this point honestly.

Author Response

Response to Reviewer 1 comments

Reviewer 1

1) General comments

The authors reported the novel genes, ARRDC4 and UBXN1, in prostate cancer. I am sure this is the first report to show an instrumental feature of these newly identified molecular in that field. And the showed an attractive profile of UBXN1 as the ubiquitin activity to BRCA1, which has been known for being one of the main gene as a hereditary breast cancer syndrome also presented as prostate cancer in men. The authors added a somewhat new prospect to the current literature; however, the study includes several points that need to be modified to qualify publication. The reviewer would like to suggest critical points to improve and increase the importance of this manuscript.

2) Specific comments:

1. No survival outcome was not analyzed despite enough number of patients and enough period after surgery. At least, biochemical recurrence free survival should be needed. Although Gleason score has been the best predictor of treatment outcome, this pathological finding cannot be associated with biological alteration directly. For instance, nuclear grade each cell can more reflect alteration of molecular.

Authors’ reply: We appreciate reviewer’s comment. As reviewer’s comments, survival outcomes which affect prognosis in real clinical situation. We additionally investigate biochemical recurrence free survival according to variants. The patients who have ARRDC4 variant (rs200944490) had adverse biochemical free recurrence outcomes, 5-year biochemical recurrence free survival rate was estimated 24.9% compared with the patients who have not 69.9% (log rank test p < 0.001). The Kaplan-meier analysis was shown in this figure (figure was uploaded in word file)

As reviewer’s comments, we newly added relevant survival outcomes in results section (line 299-302) as followings:

3.8. Biochemical recurrence free outcomes

During a median follow-up of 58 months, 130 patients had experienced biochemical recurrence (BCR). 5 year BCR free survival rate was achieved 24.9% in patients with variants and 69.9% in patients without (log rank test p < 0.001) (Figure 7)

2. At the first exploring exomic array analysis, the four SNPs were identified as risk of high Gleason score. Then, authors proceeded to functional analysis of ARRDC4 and UBXN1. To be general, since only one SNP should not affect molecular function, an effect of the SNPs must be analyzed or discussed.

Authors’ reply: We fully understand and agree reviewer’s comments. Among four SNPs, we performed functional analysis using two SNPs, due to these two SNPs had more significance than others. As reviewer’s view, there was controversy only one SNP could affect the disease phenotype. Therefore many researchers had several methods to overcome this limitation by using polygenic risk score or functional analysis. However polygenic risk score which was developed with many SNPs was not appropriate to perform functional analysis like our experiments. So we only hope reviewer’s generous understanding for the issue. We newly added relevant limitation to discussion sections (line 354-356) as followings

“The functional analysis using only one SNP could be limitations due to methodological limitation. To overcome this limitation, we needed polygenic risk score which aggregate influence of SNPs in future study”

3. Detail of tissue preparation for exomic array was inappropriate. This methodological matter must be important and describe carefully because inappropriate tissue handling might lead to wrong results. Since Gleason score in prostate cancer is very varied, even in one patient, we should only use representative Gleason score tissue for the array. Authors should show this point honestly.

Authors’ reply: We appreciate reviewer’s comment. Exome array was conducted with blood sample of included patients, not prostatectomy specimen tissue. We think this was major limitation and our future project to validate our study. The Gleason grade was made by single genitourinary pathologist through recent International Society of Urological Pathology (ISUP) recommendation of modified Gleason score. So we only hope reviewer’s generous understanding for the issue. We newly added relevant limitation to discussion sections (line 356-358) as followings :

Another limitation was we only perform exome array in blood sample, not validate our results in prostatectomy tissue.

Reviewer 2 Report

In the manuscript entitled “ARRDC4 and UBXN1: Novel Target Genes Correlated with Prostate Cancer Gleason Score”, Jong Jin Oh and colleagues, using exome array, identified four SNPs that are associated with Gleason scores of prostate cancer. Two of the SNPS are located within the functional genes ARRDC4 and UBXN1. They knocked down the expression of these genes individually and found that reduced expression of these genes resulted in decreased cell proliferation, migration, invasion and EMT, concurrent with decreased activity of PI3K/Akt/NF-κB signaling pathway. Overall, the finding of this study is novel and could contribute to our understanding on the involvement of ARRDC4 and UBXN1 in prostate cancer. 

Concerns:

  1. I recommend authors to assess the protein and mRNA levels of ARRDC4 and UBXN1 in normal prostate and prostate cancer tissues. Antibodies against these proteins are commercially available.
  2. I recommend using Western blot to confirm the knockdown efficiency of ARRDC4 and UBXN1
  3. Wound healing and invasion assays lasted for 24-48 hours. The reduced wound healing and invasion might be caused by reduced cell proliferation since knockdown of ARRDC4 and UBXN1 affected cell proliferation. I recommend reducing the experiment duration of the wound healing and invasion assays.
  4. There is a grammar error in the abstract “In this study, we found that germline variants of ARRDC4 and UBXN1 could be affect the permanent Gleason score, …”
  5. The subtitle “3.6. Knockdown of UBXN1 and ARRDC4 Induced Apoptosis and EMT” does not align with the results “Reduced expression of UBXN1 and ARRDC4 in PCa cells resulted in down-regulation of vimentin (an important mesenchymal marker), and upregulation of E-cadherin (commonly used epithelial marker) compared with negative control-siRNA- transfected cells”

Author Response

Response to Reviewer 2 comments

Reviewer 2

In the manuscript entitled “ARRDC4 and UBXN1: Novel Target Genes Correlated with Prostate Cancer Gleason Score”, Jong Jin Oh and colleagues, using exome array, identified four SNPs that are associated with Gleason scores of prostate cancer. Two of the SNPS are located within the functional genes ARRDC4 and UBXN1. They knocked down the expression of these genes individually and found that reduced expression of these genes resulted in decreased cell proliferation, migration, invasion and EMT, concurrent with decreased activity of PI3K/Akt/NF-κB signaling pathway. Overall, the finding of this study is novel and could contribute to our understanding on the involvement of ARRDC4 and UBXN1 in prostate cancer. 

Concerns:

1. I recommend authors to assess the protein and mRNA levels of ARRDC4 and UBXN1 in normal prostate and prostate cancer tissues. Antibodies against these proteins are commercially available.

Authors’ reply:  Thank you for your valuable comments. We totally agree to your comments. Unfortunately, it is difficult to obtain normal prostate tissues from patients.

ARRDC4 and UBXN1 expression in prostate tissue and cancer can be confirmed in “The Human Protein ATLAS (https://v15.proteinatlas.org)”.

(detailed response in attached word file, we can not upload image file in this text box system). 

Unfortunately, our laboratory does not have a ARRDC4 and UBXN1 antibody. threfore additional experiments you recommended are difficult to perform within the given time (10 days) since the antibody need to be received after ordering. It takes 6~8 weeks for experiments. We only hope reviewer’s generous understanding for the issue. We newly added relevant limitation in discussion section (line 358-359) as followings :

Another technical limitation which we could not performed the protein and mRNA level of ARRDC4 and UBXN1 in normal prostate should be resolved in next study.

2. I recommend using Western blot to confirm the knockdown efficiency of ARRDC4 and UBXN1

Authors’ reply:  Thank you for your concern. We totally agree to your comments.

In other studies, knockdown efficiency was also confirmed by RT-PCR and Western blot. The knockdown efficiency showed similar results in RT-PCR and Western blot (1, 2) (detailed figure file was present in uploaded word file)

Fig. 1. Effect of siRNA interference on the expression of AQP-5 in MCF-7/ADR cells. (A) The expression of AQP-5 protein detected by Western blot. (B) The expression of AQP-5 mRNA detected by RT-PCR.

Fig. 2. Si-RNA inhibited PAEP expression. (A) PAEP mRNA level of 624.38-Mel cells following siPAEP10-12 transfection, determined by RT-CR (B) qPCR and (C) western blot analysis indicates that the secreted PAEP protein levels of two melanoma cell lines were decreased following siPAEP10-12 transfection.

  1. Li et al., Onco Targets Ther 2018; 11: 3359-3368. Effect of AQP-5 silencing by siRNA interference on chemosensitivity of breast cancer cells. PMID 29922074
  2. Chai et al., Mol Med Rep 2013; 8: 1390-1396. Optimization and establishment of RNA interference-mediated knockdown of the progestagen-associated endometrial protein gene in human metastatic melanoma cell lines. PMID 24042729

Our laboratory does not have a ARRDC4 and UBXN1 antibody. Sorry, but additional experiments you recommended are difficult to perform within the given time (10 days) since the antibody need to be received after ordering. It takes 6~8 weeks for experiments. It is a good point, so if you recommend additional experiments, we will secure the antibody for western blot and used it.

3. Wound healing and invasion assays lasted for 24-48 hours. The reduced wound healing and invasion might be caused by reduced cell proliferation since knockdown of ARRDC4 and UBXN1 affected cell proliferation. I recommend reducing the experiment duration of the wound healing and invasion assays.

Authors’ reply:   Thank you for your sharp comments.

After transfection, cell seeding number for wound healing and invasion assay was same in all experiments group. Therefore, just as is the decrease in cell proliferation was the result of knockdown of ARRDC4 and UBXN1, the reduced wound healing and invasion is also thought to be an effect of knockdown of ARRDC4 and UBXN1. Similar to our finding, in a study by Sun et al (1), cell proliferation was decreased in PC3 cells transfected with MCT4 siRNAs, which persisted up to 72 h post transfection. The cell proliferation was decreased in a time-dependent manner (A). MCT4 knockdown inhibited PC3 cells invasion as measured by transwell assay. PC3 cells were grown and transfected with MCT4 siRNA for 48 h (B). (detailed figure file was present in uploaded word file)

Unlike other cancer cells, prostate cancer cells have a very slow growth rate, so cell migration is small in less than 24 hours. In other studies, LNCaP cells are observed up to 60 hours (2).

  1. EXCLI J 2019; 18: 187-194. MCT4 promotes cell proliferation and invasion of castration-resistant prostate cancer PC-3 cell line. PMID 30975102
  2. Tiwari et al., 2019; 19: 346. Reduced ERG1 expression promotes prostate cancer progression and affects prostate cancer cell migration and invasion. PMID 30975102

4. There is a grammar error in the abstract “In this study, we found that germline variants of ARRDC4 and UBXN1 could be affect the permanent Gleason score, …”

Author’s reply : we revised that sentence in simple summary (line 15)

5. The subtitle “3.6. Knockdown of UBXN1 and ARRDC4 Induced Apoptosis and EMT” does not align with the results “Reduced expression of UBXN1 and ARRDC4 in PCa cells resulted in down-regulation of vimentin (an important mesenchymal marker), and upregulation of E-cadherin (commonly used epithelial marker) compared with negative control-siRNA- transfected cells”

Author's reply :  Thank you for your comments. As you recommended, we changed “3.6. Knockdown of UBXN1 and ARRDC4 reduced Apoptosis and EMT” (Lines 263).

Round 2

Reviewer 1 Report

The authors answered earnestly for my inquiries.